# Cardiac Dysfunction in Rheumatoid Arthritis: The Role of Inflammation

**DOI:** 10.3390/cells10040881

**Published:** 2021-04-13

**Authors:** Jianmin Chen, Lucy V. Norling, Dianne Cooper

**Affiliations:** 1Centre for Biochemical Pharmacology, William Harvey Research Institute, Barts and The London School of Medicine and Dentistry, Queen Mary University of London, London EC1M 6BQ, UK; jianmin.chen@qmul.ac.uk (J.C.); l.v.norling@qmul.ac.uk (L.V.N.); 2Centre for Inflammation and Therapeutic Innovation, Queen Mary University of London, London EC1M 6BQ, UK

**Keywords:** rheumatoid arthritis, inflammation, heart failure, resolution

## Abstract

Rheumatoid arthritis is a chronic, systemic inflammatory disease that carries an increased risk of mortality due to cardiovascular disease. The link between inflammation and atherosclerotic disease is clear; however, recent evidence suggests that inflammation may also play a role in the development of nonischemic heart disease in rheumatoid arthritis (RA) patients. We consider here the link between inflammation and cardiovascular disease in the RA community with a focus on heart failure with preserved ejection fraction. The effect of current anti-inflammatory therapeutics, used to treat RA patients, on cardiovascular disease are discussed as well as whether targeting resolution of inflammation might offer an alternative strategy for tempering inflammation and subsequent inflammation-driven comorbidities in RA.

## 1. Cardiovascular Disease in Rheumatoid Arthritis (RA): Impact on the RA Population

Rheumatoid arthritis (RA) is a systemic inflammatory disease and the most common autoimmune arthritis, affecting up to 1% of the population [1]. Peak incidence occurs around age 50 and affects twice as many women than men. Although the disease is characterised by inflammation within synovial joints, which may ultimately lead to bone and cartilage destruction, extra-articular manifestations result in a 1.5-fold higher risk of death than the general population [2]. The major cause of this increased risk of premature death is disease in the cardiovascular system [3,4]. In fact, RA is recognised as an independent cardiovascular risk factor in European Society of Cardiology guidelines [5] and the European League Against Rheumatism (EULAR) recommends cardiovascular disease (CVD) risk scores, predicted by tools designed for the general population, be multiplied by 1.5 for RA patients. This allows for nontraditional risk factors, such as inflammatory status, to be taken into account [6]. The impact of clinical disease activity of RA patients on CV risk raises important questions for treatment and whether CVD risk could be reduced by tighter control of inflammation and thus disease activity in RA patients. Indeed, the introduction of earlier, more aggressive treat-to-target strategies [7] appears to be having some impact on CVD risk; a recent Swedish control case study found that patients diagnosed post-2004 had a similar CVD mortality risk to comparators, in contrast to patients diagnosed before 2003 who had significantly elevated CVD mortality [8]. This is supported by findings in population-based cohort studies from the USA and Canada, which indicate that CVD risk of patients with RA after the turn of the century has improved significantly and in the US study was largely driven by a significant decrease in incidence of myocardial infarction (MI) [9,10]. Whilst progress is clearly being made through earlier diagnosis and more effective treatment, some studies still indicate that mortality rates remain higher in the RA population, which may reflect access to and standard of care provided. The COMORA (comorbidities in rheumatoid arthritis) study assessed comorbidities in 3920 patients across 17 countries between 2011 and 2012 and identified variability in both the prevalence of comorbidities and associated risk factors across countries, which is likely due to differences in detection, management, and prevention. Importantly, it confirmed that monitoring and management of CV risk in RA patients is suboptimal [11]. Whilst it is acknowledged that cardiovascular death in RA patients is declining in some cohorts [8,9], a recent United States National Inpatient Sample study found CVD complications, such as congestive heart failure (CHF), acute myocardial infarction (AMI), and atrial fibrillation in RA patients, were significantly increased between the years 2005 and 2014, along with increasing numbers of RA patients and a greater proportion of patients with comorbidities [12]. Of interest, newly diagnosed RA patients have significantly higher event rates of stroke and heart failure in the 5 years preceding diagnosis and a 1.4-fold higher risk immediately following diagnosis. This is despite not exhibiting typical features of CV risk such as hypertension and hypercholesterolemia, indicating that CVD is not just a late complication of RA [13]. In this review, we discuss the role inflammation plays in CVD in RA with a focus on heart failure, RA therapeutics, and their ability to dampen inflammation and how this impacts CVD. We will also consider resolution of inflammation and whether therapeutics that induce pro-resolving pathways offer potential for treating CVD in RA patients.

## 2. Inflammation: The Link between RA and CVD

CVD in RA can be divided into that arising from insufficient blood supply (ischemic heart disease) occurring as a consequence of atherosclerotic disease and nonischemic disease, which occurs in the absence of coronary artery disease and is associated with changes in the cells of the cardiac muscle.

### 2.1. Ischemic Heart Disease

A role for inflammation in atherogenesis is clear and has led to trials to assess the efficacy of anti-inflammatory therapeutics on CV events [14,15,16]. Accelerated atherosclerosis in RA patients is contributed by increased systemic and vascular inflammation [17,18], and studies indicate a close correlation between ischemic heart disease, CV risk, and RA disease activity [19,20], with increased risk associated with the number of acute flares of arthritis [21]. The risk of developing coronary plaques is higher in RA patients compared with non-RA subjects [22]. Consequently, this increased coronary atherosclerosis predisposes RA patients twice as likely to experience silent AMI [23], and several studies have shown RA to be associated with a two- to four-fold higher risk for AMI [24,25] with a similar risk of MI to that associated with diabetes mellitus [26]. Indeed, a recent prospective cohort study found the incidence of CV events, which included MI, was more than double in RA patients compared with non-RA subjects and even higher than that in patients with type 2 diabetes [27]. A cross-sectional study of 1,112,676 patients with MI in the United States found that RA patients had 38% increased likelihood of undergoing thrombolysis and a 27% increased likelihood of undergoing percutaneous coronary intervention compared with other patients [27]. Moreover, patients with RA have poorer long-term outcomes indicated by higher mortality and higher incidence of recurrent ischemia compared with patients without RA [28].

### 2.2. Cardiac Dysfunction/CHF

CHF is the second highest cause of death in RA patients after myocardial infarction, with studies suggesting up to a two-fold increase in the incidence of heart failure in the RA population [29,30,31,32]. HF can arise as a result of ischemic and nonischemic disease and both types are increased in RA patients [31]. There are however some differences in susceptibility, which may be due to differences in underlying pathogenesis. Ischemic HF was found to be increased in rheumatoid factor positive patients, whereas nonischemic HF was found to increase rapidly following diagnosis and was associated with inflammatory activity and disease activity but not rheumatoid factor positivity [31]. It is likely that HF is underdiagnosed in the RA population with subclinical and asymptomatic changes occurring over many years. A recent cohort study of 355 RA patients by Ferreira et al. [33] found that almost one third of patients met the study characteristics for HF, but only 7% of the cohort had a diagnosis of HF prior to study enrolment. RA patients with HF also presented a higher prevalence of traditional CV risk factors such as diabetes and dyslipidaemia as well as higher concentrations of markers of inflammation and fibrosis, findings in-line with HF patients without RA. The evidence accumulating in the literature suggests that RA should be considered a risk factor for HF, especially in newly diagnosed patients with high disease activity [30].

HF can be phenotypically defined clinically based on left ventricular ejection fraction with patients defined as having HF with reduced ejection fraction (HFrEF; EF ≤ 40%) or HF with preserved ejection fraction (HFpEF; EF ≥ 50%) [34]. The incidence of HFpEF is on the rise, in line with an increasingly aged population, such that it is becoming the dominant form of HF, with a recent retrospective study identifying HFpEF in 64% and 62% of patients with and without RA, respectively [35]. Of interest here, patients with the highest levels of C-reactive protein (CRP) had the greatest HF risk, suggesting a role for inflammation in the pathogenesis.

Whilst treatment options for HFrEF have advanced in recent years, such that mortality is declining, the long-term prognosis for HFpEF patients remains bleak. The heterogeneity of the condition, which is associated with multiple comorbidities, underlies the complex pathophysiology of the condition. These factors, combined with a lack of robust animal models, has hindered the development of new treatments [36]. It is apparent from both human and murine studies that there is a link between comorbidities associated with systemic inflammation and the development of HFpEF [37,38,39], and that inflammation-related biomarkers are strongly associated with HFpEF severity and outcome [40].

## 3. CVD Risk Factors in the RA Population

The increased propensity of RA patients to develop CVD is multifactorial and only partially explained by an increased prevalence of traditional cardiovascular risk factors. Chronic systemic inflammation and potential side-effects from medications are also contributing factors. Studies indicate that traditional CV risk factors are often higher in the RA population than the general population, particularly smoking, obesity, and diabetes; however, they do not confer the same risk profile in RA and non-RA subjects [41,42] suggesting a “risk factor paradox”. The background of inflammation in RA patients modifies the applicability of traditional risk factors, such as lipid levels, for determining CV risk in RA, as evidenced by the so-called “lipid paradox”, whereby patients with lower levels of low-density lipoprotein cholesterol (LDL-C) carry a higher CV risk than those with higher levels [43,44,45]. Studies show that RA patients have lower levels of total cholesterol and LDL than the general population [46,47], which may be due, at least in part, to increased catabolism of cholesterol ester in RA patients [48]. The reduction in inflammation in response to treatment with disease-modifying antirheumatic drugs (DMARDs) is associated with increased circulating lipid levels in RA patients, although high-density lipoprotein (HDL) function may be improved [49]. As the link between inflammation, lipid levels, and their relative contribution to CV risk is not yet fully understood, the current therapeutic strategy remains to reduce RA disease activity through a lowering of inflammation in patients as a means to reduce CV risk [49].

### Inflammation as a CV Risk Factor

RA patients still have an increased risk of CVD, even after adjustment for traditional risk factors [24,42,50]. The link between RA disease activity, raised CRP levels, and CVD risk indicates that persistent inflammation is a key mechanism in the development of CVD [51,52]. Analysis of the Consortium of Rheumatology Researchers of North America (CORRONA) registry uncovered a significant trend towards reduced CV events with declining disease activity, even after adjusting for immunomodulatory agents [53]. The analysis of the CORRONA cohort focused on ischemic heart disease and did not include HF; however, Mantel et al. compared the incidence of ischemic versus nonischemic HF in 10,000 Swedish patients and found that risk of HF increased rapidly after RA onset and was also associated with high disease activity [31]. Increased systemic inflammation, as indicated by elevated CRP levels, is associated with increased risk of HF in RA patients, independent of traditional cardiovascular risk factors [35]. After stratifying by HF subtype, CRP levels accounted for a greater proportion of HFpEF compared with HFrEF, suggesting that inflammation may be a greater contributor to the former in RA patients [35]. Figure 1 highlights the contribution of both traditional risk factors and systemic inflammation to CVD in RA patients.

In RA patients, local inflammation in the myocardium is also evident by cardiac magnetic resonance imaging, even in a proportion of patients with no overt CVD [54,55] and correlates with the development of CHF [56]. Increased expression of the pro-inflammatory cytokines tumour necrosis factor (TNF), interleukin (IL)-1α and IL-1β, and increased expression of cell adhesion molecules in heart biopsies are further indicators of heightened local inflammation within the cardiac muscle of RA patients [57].

## 4. The Role of Inflammation in Heart Failure

In non-RA settings, it is well known that the pathology of HFpEF is linked to immune dysregulation and systemic inflammation [58]. This results in cardiac remodelling, which leads to functional changes characterized by left ventricular hypertrophy, fibrosis, and diastolic dysfunction due to impaired left ventricular relaxation. The incidence of HFpEF in other conditions associated with chronic inflammation such as obesity, diabetes mellitus, and chronic kidney disease [58] suggests that elevated circulating pro-inflammatory cytokines in RA patients are likely determinants for the pathogenesis of heart failure. Additionally, specific autoantibodies developed against antigens present in the myocardium might provoke local inflammation via their interaction with activating Fc receptors [59] and, as such, contribute to the development of heart failure.

Inflammation undoubtedly plays a role in both forms of HF albeit via different mechanisms. HFpEF is associated with higher levels of circulating biomarkers of inflammation than HFrEF, which is associated with higher levels of markers of myocyte stress and progressive loss of cardiomyocytes [38,60,61]. Systemic inflammation results in increased levels of circulating pro-inflammatory cytokines such as TNF-alpha and interleukin-6, leading to endothelial activation, nitrosative and oxidative stress, increased myocardial stiffness, and fibrosis, ultimately resulting in diastolic dysfunction as a result of impaired relaxation of the myocardium [60]. HFpEF patients have higher circulating levels of inflammatory markers than HFrEF, hypertensive patients, or those with unstable versus stable disease [62,63,64,65,66]. Endomyocardial biopsies of patients with diastolic dysfunction showed increased numbers of infiltrating leukocytes expressing transforming growth factor (TGF) β as well as increased collagen levels [67], indicating a link between inflammation and fibrosis. Furthermore, levels of inflammatory mediators can be used to stratify risk of diastolic dysfunction, with patients that cluster into an “inflammatory phenotype” having the highest risk of developing diastolic dysfunction [68]. In light of this, it is perhaps not surprising that RA patients with increased levels of disease activity and systemic inflammation have increased risk of HFpEF [69]. Assessment of cardiac muscle biopsies from patients with inflammatory rheumatic diseases, including RA, reveals inflammation within the tissue in the form of infiltrated immune cells, as well as expression of pro-inflammatory cytokines and adhesion molecules [57]. Whilst circulating cytokines are elevated in RA patients [70], the heart itself is also capable of local production from cardiac myocytes [71]. Elevated levels of inflammatory cytokines lead to endothelial dysfunction through increased generation of reactive oxygen species, which results in a decrease in nitric oxide (NO) bioavailability [72]. Endothelial activation as a result of increased levels of inflammatory mediators leads to the recruitment of immune cells, in particular monocytes, which differentiate into profibrogenic major histocompatibility complex class II (MHCII) high macrophages into the heart [73]. Figure 2 indicates the mechanisms that link systemic inflammation to HFpEF.

Mechanistic insights into the processes that link inflammation to cardiac pathology are often derived from animal studies. In the case of RA, animal models investigating cardiac function are limited. In a recent experimental study [74], systolic dysfunction in collagen antibody-induced arthritis was reported, these mice developed cardiac hypertrophy and fibrosis which were associated with increased expression of TNF locally in the heart and systemically in serum. Interestingly, TNF levels in the heart remained increased, even after overt signs of arthritis had diminished. Local cytokine production in the heart may drive cardiac remodelling, a theory supported by the development of HF in mice engineered to overexpress TNF in cardiomyocytes [75]. In a separate study [76], Zhou et al. reported susceptibility to cardiac systolic dysfunction and dilated cardiomyopathy in mice with collagen antibody-induced arthritis with enhanced cardiac inflammatory cell infiltration and increased inflammatory gene expression including TNF-α and IL-6 in isolated ventricular cardiomyocytes and cardiac fibroblasts. In adjuvant-induced arthritis in rats, evidence has been shown of cardiac hypertrophy and endothelial dysfunction in coronary arteries as well as an exacerbated response to ischemia-reperfusion injury [77]. Mokotedi et al. [78] recently reported diastolic impairment in rats subjected to collagen-induced arthritis, which correlated with increased levels of circulating inflammatory cytokines. K/BxN F1 mice, which spontaneously develop arthritis, also develop endocarditis. This endocarditis was found to be dependent on local cardiac inflammation provoked by interaction of autoantibodies and Fcγ receptors [79]. These studies support a link between systemic inflammation in arthritic animals and cardiac function and pave the way for further mechanistic studies to understand the pathogenesis of CVD in RA.

Murine models of HFpEF, per se, are less established than those for HFrEF, particularly coupled with inflammatory comorbidities. Pressure overload models in mice provide evidence of a role for inflammation in HF development, with increased levels of pro-inflammatory cytokines such as TNF-alpha, IL-1 and IL-6 produced by cardiac myocytes, and infiltration of immune cells into the myocardium [80,81,82]. Abatacept, the CTLA4-Ig fusion protein licensed for treating rheumatoid arthritis, significantly reduces cardiac dysfunction in a pressure overload model of HF in mice through inhibition of T cell and macrophage activation and subsequent fibrosis in an IL-10-dependent manner, even when administered at an advanced stage [83]. Of note, the effects of abatacept were evident at a dose range comparable to that used in RA patients. Mechanistic information can be derived from studying other comorbidities that also induce systemic inflammation. For example, in a swine model examining the influence of diabetes mellitus, hypercholesterolemia, and hypertension, plasma levels of TNF-alpha were significantly raised, and endothelial dysfunction was apparent in coronary arteries as a blunted response to the vasodilator bradykinin [84]. Uncoupling of endothelial nitric oxide synthase (eNOS) along with enhanced NADPH oxidase activity resulted in increased oxidative stress, which ultimately led to left ventricular diastolic dysfunction. A decline in NO bioavailability results in reduced cyclic guanosine monophosphate (cGMP) and protein kinase G (PKG) activity, the result of which is reduced phosphorylation of the giant intracellular protein titin [85]; hypophosphorylation of the stiff N2B titin isoform raises the resting tension of cardiomyocytes [86]. Endothelial dysfunction/activation as well as oxidative stress is also evident in rodent models of HFpEF induced by senescence and western diet, diabetes, and hypertension [85,87]. In a “two-hit” model in mice, Schiattarella et al. induced hypertension through NOS inhibition coupled with a high-fat diet to recapitulate systemic and cardiovascular features of HFpEF [88]. Cardiac hypertrophy and fibrosis were evident as well as impaired endothelial function in coronary arteries with upregulation of inflammatory cytokines both systemically and in cardiac tissue. Increased nitrosative stress occurred through increased inducible nitric oxide synthase (iNOS) levels in the left ventricles of mice, which correlated with findings in human HFpEF hearts. Further elegant experiments led to the identification of a causal role for downregulation of the cardiac specific spliced form of *Xpb1s* in HFpEF pathogenesis as a result of *S*-nitrosylation of the unfolded response protein sensor IRE1α. Endothelial dysfunction is evident in both RA patients as well as in rodent models of RA [77,89,90]. Again, this is linked to eNOS uncoupling and reduced NO bioavailability as well as increased iNOS expression/activity and increased levels of the NOS inhibitor asymmetric dimethylarginine (ADMA) [91,92,93,94].

Both cardiac macrophages and fibroblasts contribute and respond to inflammatory mediators leading to increased collagen deposition and fibrosis. Numbers of cardiac macrophages double in HFpEF patients and express increased transcript for TGF beta, a potent activator of fibroblasts [67,73]. Circulating levels of the macrophage-derived profibrotic protein galectin-3 are associated with worse outcomes for HFpEF patients; galectin-3 acts upon fibroblasts to induce proliferation and collagen production [95,96] and levels are elevated in sera and synovial fluid of both undifferentiated arthritis (pre-RA) as well as diagnosed RA patients [97,98,99]. Of importance here, levels of galectin-3 in RA patients correlate with arterial stiffness [100]. Studies indicate that galectin-3 has a role in driving RA pathogenesis as indicated by reduced severity of arthritis in galectin-3 knockout mice and induction of pro-inflammatory cytokine secretion by synovial fibroblasts [101,102]. Whether galectin-3 plays a pathogenic role in CVD in the context of RA requires further investigation, but it seems plausible that it may be involved in macrophage-fibroblast crosstalk in the heart as well as the rheumatoid joint. A thorough investigation of fibroblast and macrophage phenotype in cardiac tissue from arthritic animals is required to determine whether mechanisms identified in HFpEF are also at play in RA. A recent study in murine models of diastolic dysfunction induced by aging or hypertension has highlighted a key role for expansion of monocyte-derived macrophages in the heart in driving fibrosis and the resultant diastolic dysfunction [73]. The authors found macrophage-derived IL-10 to be an indirect activator of fibroblasts through an autocrine mechanism resulting in production of the profibrotic cytokines osteopontin and TGF-β [73]. These studies suggest that therapeutics that modulate macrophage and/or fibroblast phenotype may have a beneficial role in treating CVD in the RA population.

## 5. Effect of Current RA Therapies on CVD

The clear role for inflammation as a driving factor of disease in RA patients underpins the link with disease activity and the rationale to treat to target, with DMARDs, which dampen the inflammatory response. DMARDs are the mainstay of RA treatment and can be separated as synthetic (small chemical molecules given orally) or biological (monoclonal antibodies or receptor constructs administered parenterally) agents. Whilst the anti-inflammatory properties of biological DMARDs are obvious, the mechanism of action of the older synthetic DMARDs is still not fully characterised, although treatment with these agents is associated with reduced inflammation, as indicated by reduced CRP levels in responsive patients [103,104]. Figure 3 indicates known mechanisms of action of current therapeutics.

### 5.1. Synthetic DMARDs

Synthetic DMARDs fall into two categories: conventional synthetic (csDMARDs) and targeted synthetic (tsDMARDs). csDMARDs have been used for over 50 years and comprise methotrexate (MTX), sulphasalazine, leflunomide, gold salts, and hydroxychloroquine [105]. Notably, 25–40% of patients improve significantly with MTX monotherapy, and when combined with glucocorticoids, a remission rate similar to that of biological DMARDs can be achieved. Furthermore, MTX improves the efficacy of biological DMARDs [105,106]. tsDMARDs modulate a particular molecule in the inflammatory process such as Janus kinase (Jak) inhibitors.

Traditional DMARDs such as MTX, sulfasalazine, and hydroxychloroquine appear protective against CVD in RA. A prospective study showed that RA patients treated with MTX were associated with a 60% reduction in all-cause death and a 70% reduction in CV death [107] and a lower risk of HFpEF but not HFrEF [35]. The potential for MTX to impact CVD was assessed in the recent cardiovascular inflammation reduction trial (CIRT), which tested MTX for the reduction of atherosclerotic cardiovascular and cerebrovascular events in patients with a previous MI or multivessel coronary disease with either type 2 diabetes or metabolic syndrome. In these patients, MTX did not reduce inflammation (IL-1, IL-6, and CRP levels) or reduce CV events when compared to placebo, which is in contrast to findings in the RA population suggesting that protective effects in these patients may be due to a reduction in RA severity, rather than a direct cardioprotective role. Of note, patients in the CIRT trial had median hsCRP levels of 1.6 mg/L, which may account for the lack of an effect of MTX in this cohort [108].

The use of another csDMARD, sulfasalazine, reduced CV risk in RA patients in several studies [109,110]. Whilst antimalarials, such as hydroxychloroquine, have been associated with a lower CV risk in RA patients [111], a recent study has shown that hydroxychloroquine may predispose RA patients to HFpEF due to its cardiotoxicity [35,112,113].

### 5.2. Biological DMARDs

Biological DMARDs (bDMARDs) target either pro-inflammatory cytokines such as TNF and IL-6 or pathogenic immune cells. Although these agents target different inflammatory mediators/processes, all elicit American College of Rheumatology (ACR) 70 rates of around 20% in patients with active disease, indicating that mechanistically they may converge by inhibiting pro-inflammatory cytokine production [105]. In light of this efficacy in reducing arthritis severity and given the link between inflammation and CVD, numerous trials assessing the effect of these bDMARDs on CV risk have taken place.

Anti-TNF therapies have been found to reduce the CV risk in RA patients [114], likely as a result of an improvement in cardiac function through the lowering of coronary endothelial activation and progression of atherosclerosis. The reduction in risk of CV events in response to TNF inhibition does not necessarily extend to HF [114]. The RECOVER and RENAISSANCE trials comparing the human TNF receptor etanercept to placebo in over 2000 patients combined did not reveal any protective effect of etanercept; the smaller ATTACH trial yielded similar disappointing results for infliximab at the lower dose tested and increased risk of death or hospitalization at the higher dose of 10 mg/kg [115]. These findings have led to the US Food and Drug Administration warning against the use of TNF inhibitors in RA patients with CHF [116]. However, there are some small-scale studies indicating no detrimental effect of anti-TNF therapy and improved cardiac function (both systolic and diastolic function) [117,118]. Baniaamam et al. [119] recently performed a prospective study in 51 RA patients with echocardiography and assessments at baseline and post 6 months anti-TNF therapy in patients with moderate to high disease activity. No effect on cardiac function was observed although there was a 23% decrease in *N*-terminal probrain natriuretic peptide (NT-proBNP) after 6 months of therapy. These studies indicate that whilst anti-TNF biologics are associated with an overall decrease in risk of CV events, further studies are required to assess the effects on HF in the context of RA.

Whilst tocilizumab is effective in reducing arthritis symptoms and circulating inflammatory markers in RA patients, it has been linked to increased circulating lipid levels in RA patients [120]. That said, in a recent meta-analysis tocilizumab was not found to be associated with increased CV risk compared to TNF inhibitors [121], a conclusion supported by the recent ENTRACTE trial comparing tocilizumab to etanercept in RA patients with active disease and at least one CVD risk factor [120]. Furthermore, RA patients have a two-fold higher risk of sudden cardiac death due to prolongation of QTc interval than non-RA controls, and tocilizumab may be protective in this scenario, particularly in patients for whom CRP levels are significantly reduced by tocilizumab [122].

Abatacept functions by competing with CD28 for binding to CD80/CD86 to inhibit T-cell activation and carries a similar risk of major adverse cardiac events (MACE) to that of TNF inhibitors [121].

The use of interleukin-1 inhibitors in RA have been overshadowed by the success of TNF inhibitors in particular. In terms of CVD, interleukin-1 activity has been shown to improve left ventricular deformation/function in patients with RA [123], and data from another study of RA patients treated with biologics and small molecular DMARDs, including tocilizumab and interleukin-1 antibody, found no significant association with the risk of HF [35]. IL-1 inhibition may hold potential in patients with increased CV risk due to high systemic inflammation as indicated by an exploratory analysis of canakinumab anti-inflammatory outcome study (CANTOS), which indicates that baseline concentrations of IL-6 and CRP are independently associated with hospitalisation for HF and that treatment with canakinumab is associated with a dose-dependent trend towards a 24% lower risk of hospitalisation for HF at the highest dose of 300 mg. The most substantial risk reductions were observed in patients for whom canakinumab reduced inflammation as indicated by a lowering of hsCRP < 2 mg/L [124]. Ejection fraction data was not recorded as part of CANTOS; therefore, it is not clear whether the effects observed hold for both HFrEF and HFpEF.

The link between RA disease activity and CV risk highlights the role of inflammation in driving CV disease in RA patients. That the efficacy of DMARDs is linked to reductions in disease activity scores and inflammatory biomarkers/mediators provides further evidence that if inflammation is reduced, CVD is also reduced. This is supported by data from a Swedish cohort study indicating that the risk of acute coronary syndrome is 50% lower in patients with a good clinical response to TNF inhibitors compared to nonresponders [125]; a risk reduction similar to that observed in an earlier British study, which also identified a marked reduction in risk of MI in patients who responded to TNF inhibition [126].

### 5.3. Nonsteroidal Anti-Inflammatory Drugs (NSAIDs) and Corticosteroids

Both NSAIDs (in particular COX-2 inhibitors) and corticosteroids have been associated with increased CVD risk [114,127,128], and oral corticosteroids have been linked with an increased risk of nonischemic HF [31]. In contrast to TNF inhibitors and MTX, which have been found to be associated with a reduced risk of some specific CV events such as MI, corticosteroids were found to increase the risk of all CV events in a meta-analysis of controlled studies between 1960–2012 [114]. Del Rincon et al. [127] monitored 779 RA patients over almost a decade, 50% of whom were receiving corticosteroids at baseline. Both all-cause mortality and CV mortality were found to be associated with corticosteroid use independently of RA disease activity/severity and CV risk factors. Importantly, a link with the dose of corticosteroid was established with the risk of death increasing at higher doses (above 8 mg daily or 40 g, cumulatively). These findings suggest the lowest dose for the shortest period as a strategy to avoid excess risk. A recent population-based cohort analysis of patients with immune-mediated inflammatory diseases, including RA, found an increased incidence of all-cause CVD in patients across a range of doses of glucocorticoids with the highest hazards ratios of 1.75 and 1.76 observed for HF and AMI, respectively, even at a <5 mg daily dose of prednisolone. Risk was independent of disease activity level and duration [129]. When assessing risk of HF in the RA population, Mantel et al. [31] found that oral corticosteroid use was associated with a tripled risk of nonischemic HF in particular.

NSAIDs are used widely in the RA population [130] and CV risk varies depending on the NSAID, although this is true of both the RA and non-RA population. In a 2015 meta-analysis, NSAID usage was found to be associated with increased risk of all CV events; however, the authors propose that this may have been specifically related to the inclusion of studies assessing the cyclooxygenase-2 (COX-2)-specific inhibitor rofecoxib [114]. A Danish longitudinal cohort study involving over 17,000 RA patients and over 69,000 age- and sex-matched controls found no significant CV risk associated with NSAID use in RA patients compared to the control population; in fact, NSAID exposure was associated with a 22% increased risk compared to a 51% increased risk in non-RA patients [128]. Following on from the withdrawal of rofecoxib due to adverse CV outcomes, the safety of another COX-2 specific inhibitor, celecoxib was compared to the nonselective NSAIDs, naproxen, and ibuprofen in the Prospective Randomized Evaluation of Celecoxib Integrated Safety versus Ibuprofen or Naproxen (PRECISION) trial. The trial enrolled RA and osteoarthritis patients with either established CVD or an increased risk of development of CVD who required daily NSAID treatment for arthritis pain [131]. The trial demonstrated a comparable CV safety for celecoxib to naproxen and ibuprofen with celecoxib being noninferior to both of the nonselective NSAIDs for the primary outcome of CV death, nonfatal MI, or nonfatal stroke. Furthermore, renal and gastrointestinal events were significantly lower with celecoxib than ibuprofen and naproxen, respectively.

## 6. Is the Heightened Cardiovascular Risk a Result of Failed Resolution Pathways?

RA is now increasingly recognised as a disease arising as a result of failed resolution pathways [132,133]. Measures are required for assessing inflammation in RA patients as part of the assessment of CV risk. What is the impact of reducing inflammation on CV risk? Could assessing/targeting resolution be a better strategy?

The persistence of chronic inflammation may result not only from excess pro-inflammatory mediators but also from a failure of resolution responses. There is a growing body of evidence that the generation of fatty-acid-derived specialised pro-resolving mediators (SPMs) is negatively impacted in chronic inflammatory diseases including RA [134,135,136]. Levels of the DHA-derived mediator resolvin D3 (RvD3) are reduced in the serum of RA patients, and in mice, RVD3 levels are reduced in the joints in a model of delayed resolving arthritis compared to a self-resolving form [134]. Arthritis severity is also increased in preclinical models in mice lacking the biosynthetic enzymes for SPM generation [137]. Furthermore, a recent study from Dalli and colleagues has indicated that SPM profiles of RA patients can be linked to disease pathotype and DMARD responsiveness [135]. These findings strengthen the argument for patient stratification and may offer opportunities for prediction of CVD risk in RA patients. Activation of the G protein-coupled receptor (GPCR) *N*-formyl peptide receptor 2 (FPR2), a receptor known to play an important role in the regulation of inflammation and resolution, invokes resolution in murine models of inflammatory arthritis in response to the pro-resolving agonists annexinA1 (AnxA1) and RvD1 [136,138]. FPR2 is a promiscuous receptor, which can bind proteins, lipids, and peptides to transmit both pro-resolving and pro-inflammatory signals as detailed in a recent review by Perretti and Godson [139]. The ability of different ligands to induce specific conformational changes in FPR2 and the associated downstream signalling pathways elicited likely accounts for the divergent effects mediated via FPR2 [140]. Of relevance here, FPR2 agonists are also cardioprotective in experimental models of myocardial infarction [141,142,143]. Interestingly, the endogenous FPR2 agonist AnxA1 was first discovered as a downstream effector of the inhibitory effects of glucocorticoids on phospholipase A2 activity [144]. Given that corticosteroids, although anti-inflammatory, are generally associated with increased CV risk in the RA population, is indicative of how a deeper understanding of pro-resolving networks might facilitate generation of novel pharmacological entities. The identification of specific pro-resolving mediators and receptors facilitates the generation of “cleaner” therapeutics, which are designed to elicit their effects with fewer side effects. Future experimental studies are warranted to investigate whether AnxA1 or other specific pro-resolving mediators are cardioprotective in experimental models of arthritis. This approach shows promise as evidenced by studies investigating the efficacy of synthetic FPR2 agonists in models of cardiac injury. The majority of studies have focused on MI and its associated cardiac dysfunction but indicate the need for further research in other models of cardiac dysfunction. Qin et al. uncovered biased agonism by the dual FPR1/FPR2 agonist compound 17b, away from calcium mobilization in cardiomyocytes and towards induction of prosurvival ERK and Akt signalling pathways leading to reducing cardiac inflammation and injury induced by MI [145]. More recently, Asahina et al. described structural and functional improvements, again in a murine model of MI-induced heart failure in response to a novel FPR2 selective agonist [146]. The protective effects of another synthetic dual FPR1/FPR2 agonist, compound 43, was linked to skewing towards an M2 pro-resolving macrophage phenotype, again demonstrating cardioprotective effects [147]. Evidence for a protective effect of the endogenous agonist, AnxA1, comes from an elegant study of Ferraro et al. [148]. AnxA1 was found to induce a proangiogenic, reparative cardiac macrophage phenotype, and administration of the recombinant protein significantly improved cardiac function. The actions of FPR2 agonists in the context of CVD are indicated in Figure 3.

Evidence of efficacy against ischemic disease is also evident, with several SPMs shown to prevent atheroprogression and to promote plaque stabilisation [149,150]. The question remains whether these cardioprotective effects can be expanded to cardiomyopathies associated with RA. Interestingly, mice lacking the receptor FPR2 develop age-associated obesity and diastolic dysfunction [151]. Indications of beneficial effects of SPM in systemic inflammation-induced cardiac dysfunction is also evident from sepsis studies. FPR2 knockout mice have exacerbated cardiac dysfunction following caecal ligation and puncture in a model of polymicrobial sepsis [152], and RvE1 successfully attenuated polymicrobial sepsis-induced cardiac dysfunction and enhanced bacterial clearance [153], whilst maresin conjugates in tissue regeneration 1 (MCTR1) markedly improved cardiac function in a model of lipopolysaccharide (LPS)-induced cardiac dysfunction [154].

Evidence from elderly patients with CHF suggests that resolution pathways are defective, with compromised generation of RvD1 and reduced expression of the biosynthetic enzyme 15-lipoxygenase as well as the RvD1 receptor, GPR32 in peripheral lymphocytes [155]. Patients with symptomatic peripheral artery disease also have defective production of 15-epi-lipoxin (LX)A4 suggesting an inflammation-resolution deficit [156]. In addition, patients with cardiovascular disease have significantly lower plasma levels of RvD_n−3 DPA_, which correlated with markers of platelet and leukocyte activation [157].

The complex lipid status in RA patients, combined with the effects of DMARDs on lipid processing, had led to complications in decision making for prescribing lipid-lowering medications such as statins in the RA community even though they have a beneficial impact on RA associated CVD and EULAR recommendations state that they should be used as in the general population in RA patients [6,158]. It is recognised that statins have effects beyond lipid lowering, including anti-inflammatory, antiproliferative, and immuno-modulatory properties [159,160]. A recent double-blind placebo-controlled trial comparing atorvastatin to placebo in 116 RA patients found that atorvastatin significantly reduced disease activity, even after adjustment for MTX, and significantly improved CRP levels and erythrocyte sedimentation rate (ESR) as well as IL-6 and soluble intercellular adhesion molecule-1 (ICAM-1) levels [161]. Of interest here, the effects of statins also expand to generation of SPMs, which may prove beneficial against both CVD and RA. Statins have been demonstrated to increase production of 13-series resolvins (RvT) in the vasculature [162], which led to reduced joint disease as well as a reduction in systemic inflammation and decreased activation of monocytes, neutrophils, and platelets [163]. Importantly, they have also been shown to increase myocardial levels of the pro-resolving mediator 15-epi-LXA4 in rats [164]. In addition, 15-epi-LXA4 acts via activation of FPR2, which as discussed above has been shown to modulate cardioprotection in response to another pro-resolving agonist AnxA1 [141]. The fact that statins appear more effective in patients with higher disease activity might point to efficacy related to their anti-inflammatory, pro-resolving properties rather than lipid-lowering capacity [165].

It is interesting to note that drugs found to increase CV risk, such as COX-2 selective inhibitors and TNF-inhibitors, might also be resolution toxic [166].

## 7. Conclusions

Treatment of RA has improved over the last few decades, with increasing numbers of patients entering remission as a result of treat to target criteria. The anti-inflammatory actions of current medications are undoubtedly impacting incidence of cardiovascular disease; however, the primary goal is to treat systemic inflammation and aberrant immune mechanisms and hence reduce CV risk in an indirect manner. Even with earlier diagnosis and better treatment strategies, there are still patients who do not respond to current therapeutics or who do not remain in remission and cardiovascular disease in the RA population remains a concern in terms of both morbidity and mortality. There is therefore still a need for furthering our understanding of the mechanisms linking systemic inflammatory conditions such as RA with CVD, which will lead to new opportunities for the development of new treatments, particularly with regards to HFpEF, where treatment options are limited irrespective of the comorbidities driving pathogenesis. Whilst evidence exists of dysregulation of resolving pathways in both arthritic patients and animal models, there is currently insufficient evidence to determine whether these pathways can be linked to the development of cardiovascular disease in the RA population. We propose that further studies are required to understand the role of inflammation and its resolution in cardiovascular disease on a background of the ongoing systemic inflammation that is apparent in rheumatoid arthritis. A deeper understanding of these pathways, their regulation in patients with different RA pathotypes, responsiveness to current RA therapies, and whether this links to susceptibility to cardiovascular disease is therefore required.

## Figures and Tables

**Figure 1 cells-10-00881-f001:**
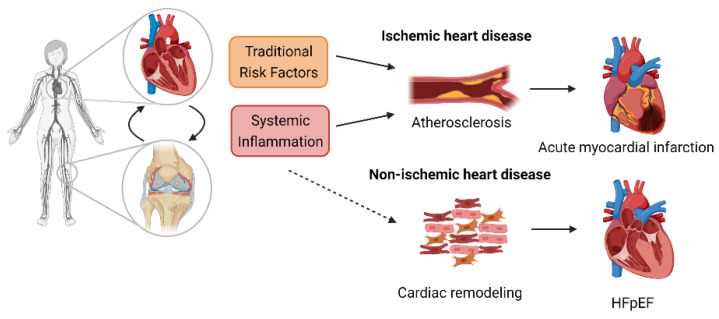
Cardiovascular disease in rheumatoid arthritis patients. Local and systemic inflammation may either directly affect the heart or make the heart more susceptible to traditional risk factors. Patients with rheumatoid arthritis (RA) have a higher susceptibility to ischemic and nonischemic heart disease. Both coronary artery disease and cardiac hypertrophy can result in heart failure.

**Figure 2 cells-10-00881-f002:**
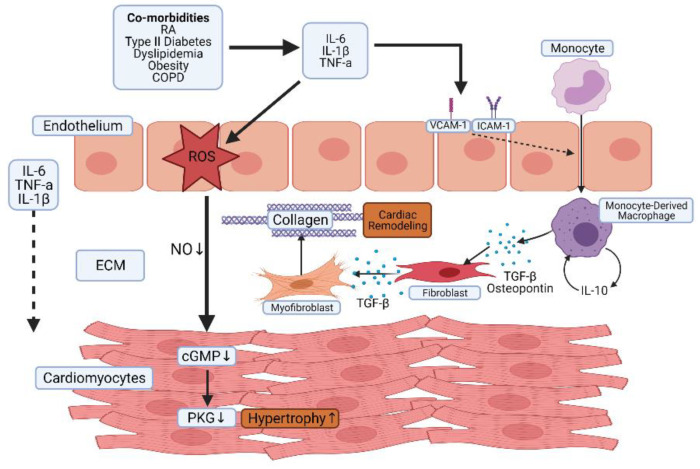
The role of systemic inflammation in the pathogenesis of heart failure (HF) with preserved ejection fraction (HFpEF). Inflammatory pathologies such as RA are associated with increased circulating levels of pro-inflammatory mediators. This results in endothelial activation and dysfunction and increased recruitment of leukocytes, such as monocytes into cardiac tissue. Increased oxidative stress contributes to a reduction in NO bioavailability and a subsequent reduction in cyclic guanosine monophosphate (cGMP) and protein kinase G (PKG) in cardiac muscle leading to cardiac hypertrophy and increased resting tension. MHCII high macrophages produce osteopontin and TGF-β in response to autocrine stimulation by IL-10, which results in fibroblast proliferation and elevated collagen deposition. Ultimately, this results in increased stiffness and diastolic dysfunction.

**Figure 3 cells-10-00881-f003:**
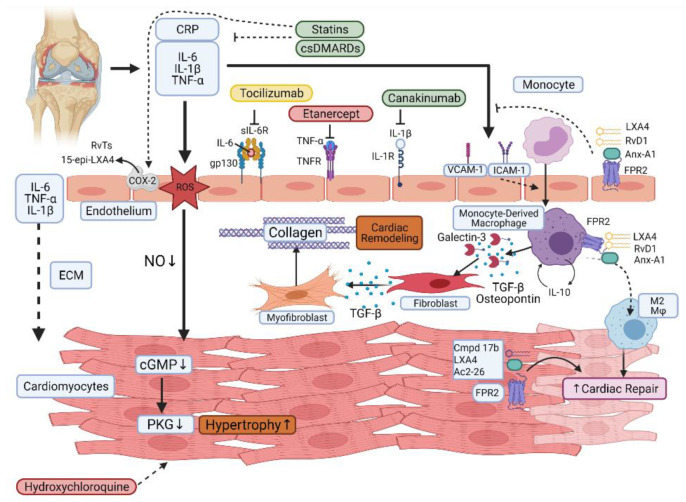
Actions of current therapeutics and FPR2 agonists in the context of cardiovascular disease in rheumatoid arthritis patients. The mechanisms of action of current therapeutics in RA patients are indicated and colour coded in relation to their cardiovascular disease (CVD) risk with green being protective and red detrimental. Actions of pro-resolving mediators LXA4 (lipoxin A4), resolvin D1 (RvD1), annexin A1 (Anx-A1), and the synthetic agonist compound 17b (Cmpd17b) mediated through the G-protein coupled receptor FPR2 are also indicated. The lipid-lowering medications statins are also included, and their impact on C-reactive protein (CRP) is indicated, as is their role in the generation of 13-series resolvins (RvT) and 15-epi-LXA4.

## Data Availability

Data sharing not applicable. No new data were created or analyzed in this study. Data sharing is not applicable to this article.

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
