# Peer review of "Cardiac Dysfunction in Rheumatoid Arthritis: The Role of Inflammation"

_cells, 2021, doi:10.3390/cells10040881_

Round 1

Reviewer 1 Report

The manuscript by Chen et al. entitled “Cardiac dysfunction in rheumatoid arthritis: the role of inflammation” presents a comprehensive overview of the current understanding of the role of inflammation in driving increased cardiovascular risk in patients with rheumatoid arthritis (RA).  Particular emphasis is placed on non-ischemic heart failure and the role of anti-inflammatory therapeutics.  The review is balanced and well-written and puts forth a novel perspective towards the end of the article that suggests new avenues in targeting the resolution of inflammation to mitigate cardiovascular risk in patients with RA. I have just a few comments that they authors may want to address.

Major points:

  1. While most of the review is comprehensive and insightful, the section of NSAIDs and corticosteroids is very brief.It may be helpful to readers to expand on the details of some of the clinical studies as was done for the preceding section on TNF and IL-6. Given that a discussion of resolution follows, it may also be helpful to describe the connection between corticosteroids and pro-resolving mediators such as annexin A1, drawing a distinction between potential therapeutic differences in these approaches.

  1. Figure 2 is very helpful in depicting the contribution of inflammation to HF.It may be helpful to add a third figure that illustrates different therapeutic approaches and how that modifies each of the depicted events.  It would be particularly helpful to add some of the pro-resolving roles, including cellular targets.

Minor points:

  1. The abbreviation “RF” is used a couple of time on page 3. I assume the authors meant to say “RA”. This should be checked.

  1. Other abbreviations, such as NT-proBNP (page 11) and ACR70 (page 10) are not clearly defined in the text.

Author Response

Thank you very much for taking the time to read our review article. We have addressed the comments raised as indicated below. 

Major points:

  1. While most of the review is comprehensive and insightful, the section of NSAIDs and corticosteroids is very brief.It may be helpful to readers to expand on the details of some of the clinical studies as was done for the preceding section on TNF and IL-6. Given that a discussion of resolution follows, it may also be helpful to describe the connection between corticosteroids and pro-resolving mediators such as annexin A1, drawing a distinction between potential therapeutic differences in these approaches.

Response 1: Thank you for highlighting these issues. We have extended the section on NSAIDs and corticosteroids on page 14 as indicated below:

Nonsteroidal anti-inflammatory drugs (NSAIDs) and Corticosteroids

Both NSAIDs (in particular COX-2 inhibitors) and corticosteroids have been associated with increased CVD risk [114,127,128] and oral corticosteroids have been linked with an increased risk of non-ischemic HF [31]. In contrast to TNF inhibitors and MTX, which have been found to be associated with a reduced risk of some specific CV events such as MI, corticosteroids were found to increase the risk of all CV events in a meta-analysis of controlled studies between 1960-2012 [114]. Del Rincon et al [127] monitored 779 RA patients over almost a decade, 50% of whom were receiving corticosteroids at baseline. Both all-cause mortality and CV mortality were found to be associated with corticosteroid use independently of RA disease activity/severity and CV risk factors. Importantly, a link with dose of corticosteroid was established with the risk of death increasing at higher doses (above 8mg daily or 40g cumulatively) findings that suggest the lowest dose for the shortest period as a strategy to avoid excess risk. A recent population-based cohort analysis of patients with immune-mediated inflammatory diseases, including RA, found an increased incidence of all-cause CVD in patients across a range of doses of glucocorticoids with the highest hazards ratios of 1.75 and 1.76 observed for HF and AMI respectively even at a <5mg daily dose of prednisolone. Risk was independent of disease activity level and duration [129]. When assessing risk of HF in the RA population, Mantel et al [31] found that oral corticosteroid use was associated with a tripled risk of non-ischemic HF in particular.

NSAIDs are used widely in the RA population [130] and CV risk varies depending on the NSAID, although this is true of both the RA and non-RA population. In a 2015 meta-analysis, NSAID usage was found to be associated with increased risk of all CV events, however the authors propose that this may have been specifically related to the inclusion of studies assessing the cyclooxygenase-2 (COX-2)-specific inhibitor rofecoxib [114]. A Danish longitudinal cohort study involving over 17,000 RA patients and over 69,000 age and sex matched controls found no significant CV risk associated with NSAID use in RA patients compared to the control population, in fact NSAID exposure was associated with a 22% increased risk compared to a 51% increased risk in non-RA patients [128]. Following on from the withdrawal of rofecoxib due to adverse CV outcomes, the safety of another COX-2 specific inhibitor, celecoxib was compared to the nonselective NSAIDs, naproxen and ibuprofen in the Prospective Randomized Evaluation of Celecoxib Integrated Safety versus Ibuprofen or Naproxen (PRECISION) trial. The trial enrolled RA and osteoarthritis patients with either established CVD or an increased risk of development of CVD who required daily NSAID treatment for arthritis pain [131]. The trial demonstrated a comparable CV safety for celecoxib to naproxen and ibuprofen with celecoxib being noninferior to both of the nonselective NSAIDs for the primary outcome of CV death, non-fatal MI or non-fatal stroke. Furthermore, renal and gastrointestinal events were significantly lower with celecoxib, than ibuprofen and naproxen respectively.

We have also expanded the section on the role of AnxA1 in the resolution section on page 15 as indicated below:

Activation of the G protein-coupled receptor (GPCR) N-formyl peptide receptor 2 (FPR2), a receptor known to play an important role in the regulation of inflammation and resolution invokes resolution in murine models of inflammatory arthritis in response to the pro-resolving agonists annexinA1 (AnxA1) and RvD1 [136,138]. FPR2 is a promiscuous receptor, which can bind proteins, lipids and peptides to transmit both pro-resolving and pro-inflammatory signals as detailed in a recent review by Perretti and Godson [139]. The ability of different ligands to induce specific conformational changes in FPR2 and the associated downstream signaling pathways elicited likely accounts for the divergent effects mediated via FPR2 [140]. Of relevance here, FPR2 agonists are also cardioprotective in experimental models of myocardial infarction [141-143]. Interestingly, the endogenous FPR2 agonist AnxA1, was first discovered as a downstream effector of the inhibitory effects of glucocorticoids on phospholipase A2 activity [144]. Given that corticosteroids, although anti-inflammatory, are generally associated with increased CV risk in the RA population, is indicative of how a deeper understanding of pro-resolving networks might facilitate generation of novel pharmacological entities. The identification of specific pro-resolving mediators and receptors will facilitate the generation of ‘cleaner’ therapeutics, which are designed to elicit their effects with fewer side-effects. Future experimental studies are warranted to investigate whether AnxA1 or other specific pro-resolving mediators are cardio-protective in experimental models of arthritis. This approach shows promise as evidenced by studies investigating the efficacy of synthetic FPR2 agonists in models of cardiac injury. The majority of studies have focused on MI and its associated cardiac dysfunction but indicate the need for further research in other models of cardiac dysfunction. Qin et al uncovered biased agonism by the dual FPR1/FPR2 agonist Compound 17b, away from calcium mobilization in cardiomyocytes and towards induction of pro-survival ERK and Akt signalling pathways leading to reducing cardiac inflammation and injury induced by MI [145]. More recently, Asahina et al described structural and functional improvements, again in a murine model of MI-induced heart failure in response to a novel FPR2 selective agonist [146]. The protective effects of another synthetic dual FPR1/FPR2 agonist, Compound 43, was linked to skewing towards an M2 pro-resolving macrophage phenotype, again demonstrating cardioprotective effects [147].  Evidence for a protective effect of the endogenous agonist, AnxA1, comes from an elegant study of Ferraro et al [148]. AnxA1 was found to induce a pro-angiogenic, reparative cardiac macrophage phenotype and administration of the recombinant protein significantly improved cardiac function.

  1. Figure 2 is very helpful in depicting the contribution of inflammation to HF.It may be helpful to add a third figure that illustrates different therapeutic approaches and how that modifies each of the depicted events.  It would be particularly helpful to add some of the pro-resolving roles, including cellular targets.

Response 2: Once again, thank you for this suggestion. We have added a third figure to indicate the mechanism of current RA therapeutics as well as the actions of pro-resolving mediators, in particular FPR agonists. 

Minor points:

  1. The abbreviation “RF” is used a couple of time on page 3. I assume the authors meant to say “RA”. This should be checked.

Thank you for indicating this oversight. We have now written rheumatoid factor in full and have added an abbreviations list as suggested by reviewer 2. 

  1. Other abbreviations, such as NT-proBNP (page 11) and ACR70 (page 10) are not clearly defined in the text.

Thank you, we have now defined both of these abbreviations in the text. 

Reviewer 2 Report

In this review article, Chen et al. summarize current knowledge on the role of inflammation as apparent in RA for cardiac dysfuntion. The article is very well written, and addresses all important aspects of the interplay of RA-associated inflammation and (resulting) CV. Therefore, only a fewmior points should be addressed:

1. The authors may include a list of abbreviations. (If I´m not mistaken, some abbreviations like "AMI" need to be explained.)

2. The authors are encouraged to include a scheme to illustrate the mode of action of DMARDS.

Author Response

We would like to thank the reviewer for taking the time to read our review article. We have addressed the comments raised as indicated below. 

1. The authors may include a list of abbreviations. (If I´m not mistaken, some abbreviations like "AMI" need to be explained.)

Response 1: Thank you for this suggestion. We have been through the manuscript to double check that all abbreviations are written in full at first use. We have also added an abbreviations list at the beginning of the review as suggested. 

2. The authors are encouraged to include a scheme to illustrate the mode of action of DMARDS.

Response 2: Thank you for this suggestion. We have added a third figure to the manuscript in which we have indicated the mechanism of action of both csDMARDs and biologics. 

Reviewer 3 Report

This review focused on the role of inflammation in RA population with CVD and the therapeutics that dampen inflammation. This paper is very well organized and clearly demonstrated solid evidence to support the author's conclusions.  This review is important for RA studies as well as CVD associated with inflammation. 

Author Response

We would like to thank the reviewer for taking the time to read our review article and for the positive feedback.